# Mining Your Memory: Client-to-Client Data Stealing in Federated Diffusion Model through Memorization

## Abstract

Federated diffusion has emerged as a promising framework for collaboratively training generative models without sharing private training data. However, we reveal a realistic and critical privacy threat of this framework: a single malicious client can steal a large portion of other clients' private training images without access to any privileged information or interfering the training process. We propose a memorization-guided data stealing attack to expose this vulnerability. This attack exploits the fact that the global diffusion model tends to memorize private training images from all clients and replicate them during generation. Based on this, a malicious client has the potential to steal private images from other clients by generating images from the global diffusion model. However, directly using the global diffusion model's default generation process rarely produces memorized samples. Therefore, we design two guidance mechanisms that significantly raise the chance of generating memorized training images of benign clients. Experiments show that by employing our attack method, an attacker can steal tens of percents of private images from other clients, while all previous data stealing attacks failed to steal any. More seriously, since our method works entirely after the federated training process, it is naturally stealthy and impossible to be detected.

## 1 Introduction

Federated learning (FL) has long been adopted to enable privacy-preserving training of discriminative models (Kairouz et al., 2021; McMahan et al., 2017). With the rapid progress of diffusion models (Ho et al., 2020; Song et al., 2021), the scope of FL has gradually expanded to generative settings, giving rise to federated diffusion frameworks that enable collaborative training of high-quality generative models without sharing raw data (Stanley Jothiraj & Mashhadi, 2024; Tun et al., 2023; Huang et al., 2024b). However, *does such explicit data isolation truly eliminate privacy leakage?* Our answer is **NO**. More seriously, we reveal that a single malicious client can steal a large portion of other clients' training data **without having access to private information from others or interfering with the normal federated training process**—it only relies on the information a client legitimately receives. To the best of our knowledge, we are the first to investigate this critical but overlooked client-to-client data stealing vulnerability in federated diffusion models, which provides a new perspective on the security of federated diffusion models.

The client-to-client data-stealing threat we identify is highly significant in practice because it possesses two properties that make it both realistic and severe. First, it originates from the client-side rather than the server. While servers in federated learning are centralized and typically well protected, clients are decentralized and often lack strict oversight; client-side attacks are therefore more frequently observed and harder to detect in real deployments compared with prior works that assumes server-side adversaries (Zhu et al., 2019; Zhao et al., 2019; Jeon et al., 2021; Fang et al., 2023). Second, the attacker has no extra privileges and does not interfere with the federated training process. Unlike inversion-based attacks (Du et al., 2025; Fang et al., 2024) that assume a scenario where the attacker has access to sensitive information from victims such as gradients (Zhu et al., 2019; Zhao et al., 2019; Fang et al., 2023) or classification labels (Zhang et al., 2020; Qiu et al., 2024; Wu et al., 2024), in our threat model the attacker has only what any legitimate federated client naturally receives during training (e.g., its own local data and the periodically updated global

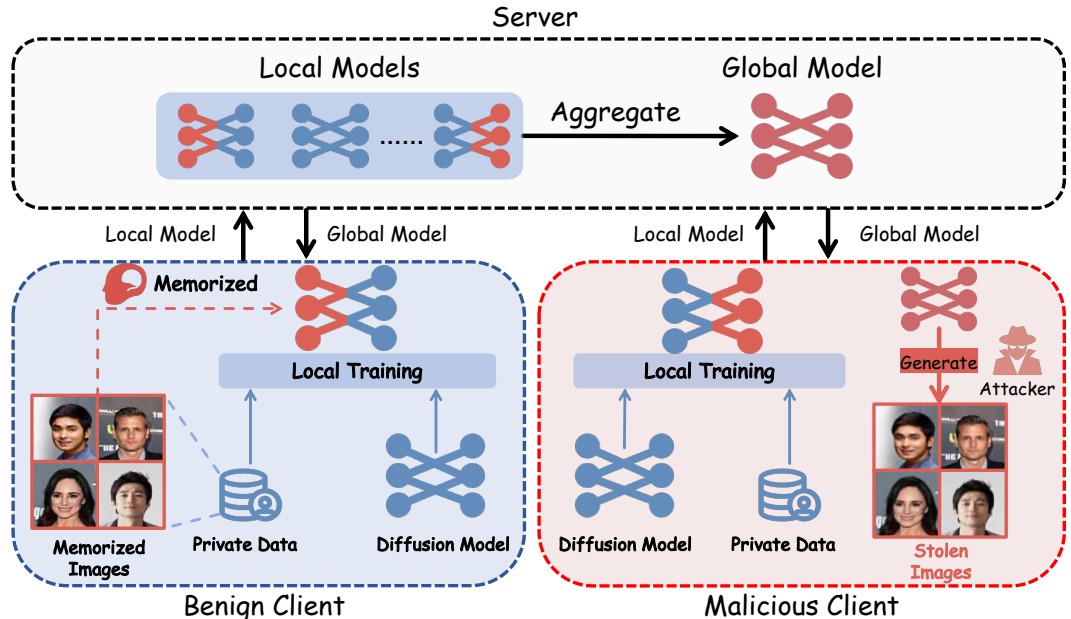

Figure 1: **Overview of the proposed threat**. During federated diffusion training, private images are memorized by local models. After aggregation, the global model implicitly retains the memorization of images from all clients, enabling a malicious client to generate private training images of others.

model). This makes the threat much easier to realize in practice and essentially undetectable under standard auditing, which greatly amplifies both its stealthiness and potential impact.

Under this realistic scenario, performing the successful client-to-client data stealing attack is technically challenging. In fact, we find that existing data-stealing methods are all infeasible in this setting. Specifically, these methods rely on external information, such as other clients' gradients (Zhao et al., 2019; Fang et al., 2023) or classification labels of private training images (Fang et al., 2023; Wu et al., 2024), which is inaccessible to the client-side attacker. This inaccessibility makes those methods inherently ineffective. Different from existing methods, we achieve the data-stealing attack by leveraging the **memorization behavior** of diffusion models. This behavior indicates that diffusion models can naturally memorize and replicate their training images during generation (Gu et al., 2025; Yoon et al., 2023; Carlini et al., 2023; Somepalli et al., 2023). Exploiting this intrinsic property of diffusion model makes it possible for the attacker to steal training images from other benign clients without any extra information.

However, we observe that using the global diffusion model's default generation process rarely yields memorized training images from other clients, making it very ineffective for data stealing. To overcome this ineffectiveness, we propose two guidance mechanisms that prioritize the generation of *memorized* images from *benign* clients and thereby significantly improving attack successful rate. The first guidance is **Threat-Focused Guidance (TFG)**. It uses a private diffusion model trained on the attacker's own data to provide negative guidance during sampling. This guidance steers generation trajectory away from the attacker's distribution and toward that of benign clients. The second guidance is **Memorization-Focused Guidance (MFG)**. It exploits the contrast between later-stage and earlier-stage global models. The earlier-stage global model mainly captures general semantics without memorizing specific samples, while later-stage models has greater potential to reproduce memorized images compared to the earlier-stage one. By contrasting their predictions, MFG suppresses non-memorized generations and amplifies memorized ones, thereby improving recovery of private data. Together, these two mechanisms enable the attacker to recover a substantial fraction of other clients' training data.

Experimental results show that the proposed TFG and MFG significantly improve the success rate of data stealing. For example, on AFHQ-Dog (Choi et al., 2020) the recovery rate increases from

4.12% to 25.02%, and on CelebA Liu et al. (2015) from 13.20% to 38.25%. In contrast, existing inversion-based methods (Fang et al., 2023; Wu et al., 2024) fail to recover any meaningful data under the same conditions.

Overall, our main contributions are summarized as follows:

- To the best of our knowledge, we are the first to investigate the client-to-client privacy leakage risk of federated diffusion models, aiming to raise awareness of this critical threat.
- We propose a novel memorization-guided attack, which leverages the memorization behavior of diffusion models to effectively perform client-to-client data stealing attack.
- Extensive experiments demonstrate that our method achieves effective data stealing in federated diffusion settings where existing inversion-based attacks entirely fail, while remaining fundamentally undetectable during training.

## 2 RELATED WORKS

### 2.1 DIFFUSION MEMORIZATION

The memorization phenomenon refers to the tendency of diffusion models to generate images that are nearly identical to those in the training set. Pioneering work (Gu et al., 2025) systematically investigates memorization of unconditional EDM (Karras et al., 2022) with different hyperparameters and training strategies. As for text-to-image generation, some works (Carlini et al., 2023; Somepalli et al., 2023; Wen et al., 2024) investigate the memorization in text-to-image diffusion models caused by unique prompts. Meanwhile, some works (Wang et al., 2024; Yoon et al., 2023; Chavhan et al., 2024) investigate memorization in diffusion models from the theoretical perspective. Based on these findings, several works aim to mitigate memorization. Some of them achieve memorization mitigation by designing different training strategies. AmbientDiffusion (Daras et al., 2023) prevents diffusion from memorizing training data by training it with noisy images, IET-AGC (Liu et al., 2024; Guan et al., 2025) neglect easy-to-remember images during training. Some of them mitigate memorization in text-to-image generation during sampling process (Wen et al., 2024; Chen et al., 2025a; Jain et al., 2025; Chen et al., 2024) by modifying the text embeddings of the prompt or the guidance scale during generation, while (Ren et al., 2024; Chen et al., 2025b) manipulate the attention maps of the text-to-image diffusion models. Other works (Hintersdorf et al., 2024; Dutt et al., 2025) eliminate the model parameters that cause the memorization. As for privacy attacks, most works leverage memorization for membership-inference attack (Ma et al., 2024; Matsumoto et al., 2023; Li et al., 2024; Pang & Wang, 2025; Jiang et al., 2025). Different from them, we leverage the memorization to directly steal private training images, uncovering a more severe privacy leakage.

### 2.2 INVERSION ATTACKS IN FEDERATED LEARNING

Inversion-based attacks aim to reconstruct the private training images of victim clients by exploiting shared information. Existing methods can be broadly categorized into Gradient Inversion Attacks (GIA) (Du et al., 2025) and Model Inversion Attacks (MIA) (Fang et al., 2024). GIA assumes that the server-side attacker who has the access to victim's gradients during training and optimizes dummy inputs to match these gradients (Zhu et al., 2019; Zhao et al., 2019). In contrast, MIA performs post-training attacks by optimizing inputs to match prediction logits of the target model (Zhang et al., 2020; Chen et al., 2021). Recent works have enhanced inversion-based attacks by incorporating generative priors or designing specialized loss objectives to improve reconstruction quality. For example, methods such as GIAS (Jeon et al., 2021) and GradInversion (Yin et al., 2021) leverage pre-trained generative models as priors to achieve efficient and high-quality reconstruction. GIFD (Fang et al., 2023) performs feature-domain inversion to align intermediate representations rather than raw pixel values. Mjölnir (Liu et al., 2025) introduces adaptive diffusion-based priors to circumvent gradient obfuscation defenses. In the model inversion domain, GMI (Zhang et al., 2020) and Deep-MIA (Khosravy et al., 2022) utilize GANs or VAEs to reconstruct training data by matching output logits or internal features. Other techniques such as VMI (Wang et al., 2021), KED-MI (Chen et al., 2021), and PLG-MI (Yuan et al., 2023) further improve inversion fidelity through variational inference or pseudo-label guided supervision. While these methods enhance attack success against discriminative models, they all fundamentally rely on semantically structured

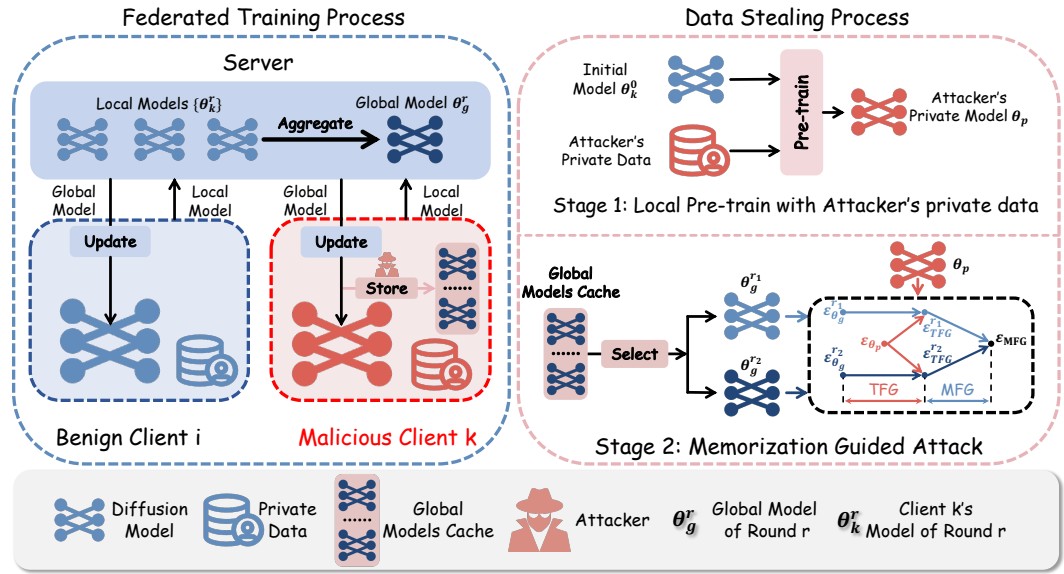

Figure 2: **Overview of our proposed memorization-guided attack.** The left panel illustrates the federated training process. Unlike benign clients, the malicious client stores global models $\{\theta_g^r\}_{r=1}^R$ across training rounds, where $R$ indicates the final training round. The right panel shows the two-stage attack: (1) the attacker pre-trains a private model $\theta_p$ with its local data; (2) the attacker selects two global models $\theta_g^{r_1}$, $\theta_g^{r_2}$ and combines them with $\theta_p$ to guide the generation process to perform the attack. $\epsilon$ indicates the noise-prediction of each model.

outputs such as classification logits or features. As a result, they are inherently incompatible with diffusion models.

## 3 METHOD

### 3.1 ATTACK SCENARIO

Our attack scenario assumes a standard federated learning (FL) setup involving $K$ clients, each holding a private dataset $\mathcal{D}_i$, where $1 \leq i \leq K$, remains local throughout training. The training process consists of $R$ communication rounds (McMahan et al., 2017). In each round $r$, where $1 \leq r \leq R$, the server derives the global model $\theta_g^r$ first and then broadcasts it to all clients, then client $i$ performs local training on $\theta_g^r$ with its private data to obtain updated parameters $\theta_i^r$, which are then sent back to the server for aggregation. The server combines these updates to produce the next global model $\theta_g^{r+1}$. After $R$ rounds of training, the final global model $\theta_g^R$ is obtained as the output of the federated process.

**Attacker's goal.** The attacker is a common client in the federated system who wants to steal training data from other clients. Suppose that the attacker's client ID is $A$, where $1 \leq A \leq K$. The attacker aims to generate a set of images $\mathcal{I}$ from the global diffusion model such that as many images as possible match those in the union of all other clients' private datasets, denoted as $\mathcal{D}_B = \bigcup_{i=1, i \neq A}^K \mathcal{D}_i$. The attack objective is to maximize the number of successfully stolen images, formally defined as $\max_{\mathcal{I}} |\mathcal{I} \cap \mathcal{D}_B|$.

**Attacker's abilities.** The attacker is an honest-but-curious client who possesses the same capabilities as any normal clients in federated system, including full access to its own private data $\mathcal{D}_A$, local training process, and the global diffusion model $\theta_g^r$ at each communication round $r$. Importantly, the attacker does not need to know system-specific configurations such as the total number of clients, nor interfere with the training procedure. This makes the attack stealthy and difficult to detect.

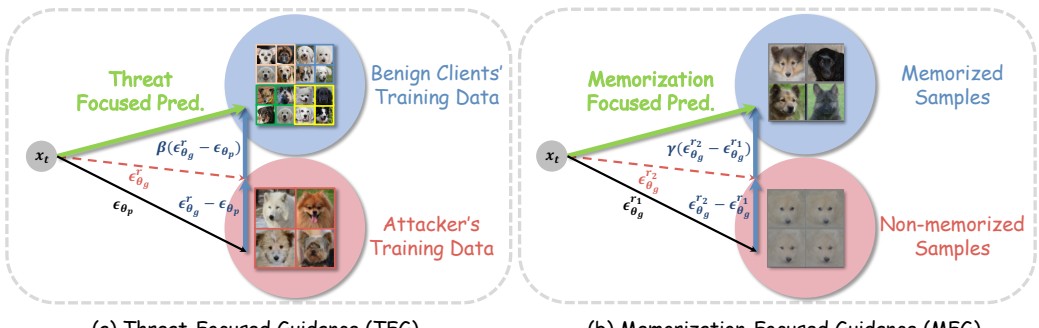

(a) Threat-Focused Guidance (TFG)    (b) Memorization-Focused Guidance (MFG)

Figure 3: Illustration of the (a) Threat-Focused Guidance (TFG) and (b) Memorization-Focused Guidance (MFG), **demonstrating their correction effect when the model's prediction $\epsilon$ deviates from the desired distribution during sampling.** In (a), images with different edge colors represent different clients, where $\epsilon_{\theta_g}^r$ is the global model's prediction that induces the attacker's private images and $\epsilon_{\theta_p}$ is the attacker's private model prediction; in (b), $\epsilon_{\theta_g}^{r_2}$ and $\epsilon_{\theta_g}^{r_1}$ are predictions from later- and earlier-round global models, corresponding to memorized and non-memorized samples, whose difference corrects the trajectory toward memorized data.

### 3.2 MEMORIZATION GUIDED ATTACK

As discussed in Sec. 1, the global diffusion model $\theta_g^r$ often memorizes private training images from all participating clients due to memorization. This phenomenon ensures the possibility for a malicious client to steal private data from others by using $\theta_g^r$ to generate memorized images.

The overall workflow of our method is shown in Fig. 2. During federated training, an honest-but-curious attacker legitimately collects the global model at each communication round, forming a cache $\{\theta_g^r\}_{r=1}^R$, and also trains a private diffusion model $\theta_p$ on its own local data with the same architecture as the global model. The cached models $\{\theta_g^r\}_{r=1}^R$ represent a sequence of increasingly powerful generators as training progresses, while $\theta_p$ characterizes the distribution of the attacker's own training data. Our two guidance mechanisms **Threat-Focused Guidance (TFG)** and **Memorization-Focused Guidance (MFG)** fully exploit the rich and complementary information embedded in $\{\theta_g^r\}_{r=1}^R$ and $\theta_p$ to steer generation toward images *memorized* from other clients, maximizing the recovery of *other* clients' private data. Details of these mechanisms are described below.

#### 3.2.1 THREAT-FOCUSED GUIDANCE

The global diffusion model $\theta_g^r$ is trained collaboratively on data from all clients, including both the attacker's private dataset $\mathcal{D}_A$ and the private datasets of other clients, denoted as $\mathcal{D}_B$. As a result, as shown in Fig. 3 (a), $\theta_g^r$ often yields prediction $\epsilon_{\theta_g}^r$ which induces the generation of memorized images from $\mathcal{D}_A$, which are useless for the attacker. Therefore, if the generation trajectory is only determined by $\epsilon_{\theta_g}^r$, the resulting image set $\mathcal{I}$ will contain a large number of memorized images originating from $\mathcal{D}_A$, thereby reducing the density of desired samples in $\mathcal{I}$.

To increase the proportion of memorized images from $\mathcal{D}_B$ in $\mathcal{I}$, the attacker needs to redirect the generation trajectory when $\epsilon_{\theta_g}^r$ tends to generate images from $\mathcal{D}_A$. To achieve this, the attacker can firstly train a private diffusion model $\theta_p$ exclusively on its own private data, and subtract the original prediction $\epsilon_{\theta_g}^r$ with $\theta_p$'s prediction $\epsilon_{\theta_p}$. This subtracted vector (illustrated as the dark blue arrow in Fig. 3 (a)) provides the model with a guidance to steer the generation trajectory away from the distribution of $\mathcal{D}_A$. As a result, the attacker increases the chance of reproducing memorized images from $\mathcal{D}_B$ and improves the density of desired images in generated images. We term this kind of guidance as the Threat-Focused Guidance (TFG).

Formally speaking, this process can be formulated as,

$$\epsilon_{\text{TFG}}(x_t, t) = \epsilon_{\theta_g}^r(x_t, t) + \beta \cdot \left( \epsilon_{\theta_g}^r(x_t, t) - \epsilon_{\theta_p}(x_t, t) \right), \tag{1}$$

---

**Algorithm 1** Memorization Guided Attack

---

**Input:** Global models $\{\theta_g^1, \theta_g^2, \ldots, \theta_g^R\}$, private dataset $\mathcal{D}_A$, guidance scales $\beta$, $\gamma$, number of generated images $N$, DDIM denoising function $\Phi(x_t, t, \epsilon)$ Song et al. (2021).
**Output:** Stolen image set $\mathcal{I}$
1: Train private diffusion model $\theta_p$ on $\mathcal{D}_A$
2: Initialize stolen image set $\mathcal{I} \leftarrow \emptyset$
3: **for** $i = 1$ to $N$ **do**
4:     Sample Gaussian noise $x_T \sim \mathcal{N}(0, I)$
5:     Initialize timestep $t \leftarrow T$
6:     **while** $t > 0$ **do**
7:         Select two global models $\theta_g^{r_1}, \theta_g^{r_2}$ with $r_2 > r_1$
8:         $\epsilon_{\text{TFG}}^{r_1} \leftarrow \epsilon_{\theta_g^{r_1}}(x_t, t) + \beta \cdot (\epsilon_{\theta_g^{r_1}}(x_t, t) - \epsilon_{\theta_p}(x_t, t))$                ▷ Eq.1.
9:         $\epsilon_{\text{TFG}}^{r_2} \leftarrow \epsilon_{\theta_g^{r_2}}(x_t, t) + \beta \cdot (\epsilon_{\theta_g^{r_2}}(x_t, t) - \epsilon_{\theta_p}(x_t, t))$                ▷ Eq.1.
10:        $\epsilon_{\text{MFG}}(x_t, t) \leftarrow \epsilon_{\text{TFG}}^{r_2} + \gamma \cdot (\epsilon_{\text{TFG}}^{r_2} - \epsilon_{\text{TFG}}^{r_1})$                      ▷ Eq.2.
11:        $x_{t-1} \leftarrow \Phi(x_t, t, \epsilon_{\text{MFG}}(x_t, t))$                                  ▷ Denoising step
12:        $t \leftarrow t - 1$
13:     **end while**
14:     Add final $x_0$ to $\mathcal{I}$
15: **end for**
16: **return** $\mathcal{I}$

---

where $x_t$ denotes the noisy image at timestep $t$ during the sampling process, $\epsilon_{\theta_g^r}(x_t, t)$ and $\epsilon_{\theta_p}(x_t, t)$ represent the noise predictions made by the global model $\theta_g^r$ and the attacker's private model $\theta_p$, respectively. The TFG scalar $\beta \geq 0$ is the guidance scale that controls the strength of the influence from the private model. The resulting noise prediction $\epsilon_{\text{TFG}}(x_t, t)$, depicted as the dark green arrow in Fig. 3 (a), is then used to guide the sampling step.

### 3.2.2 MEMORIZATION-FOCUSED GUIDANCE

We observe that although the global model memorizes many samples from $\mathcal{D}_B$ during training, its default sampling strategy still yields a substantial fraction of non-memorized outputs, which dilutes attack effectiveness. In Fig. 3 (b), the noise prediction of the global model at round $r_2$, frequently points toward these non-memorized regions and produces outputs uninformative to the attacker. To counteract this, we design a strategy that actively suppresses non-memorized generations and amplifies memorized ones.

Similar to TFG, the core idea is to use a reference model that captures the dataset's global semantics while avoiding instance-level memorization; such a model provides an in-domain negative reference that steers sampling away from the non-memorized distribution without pushing generations off the training data manifold. Prior work shows that models from early training rounds naturally satisfy these properties (Gu et al., 2025), so we choose an earlier-stage global model $\theta_g^{r_1}$ (typically from the first 10%–20% of rounds) as this reference. As illustrated by the non-memorized images in Fig. 3 (b), its outputs appear generic and blurred, reflecting broad semantics rather than specific samples. By contrasting the later-round prediction $\epsilon_{\theta_g^{r_2}}$ with the early-round prediction $\epsilon_{\theta_g^{r_1}}$ (dark blue arrow), we obtain a guidance direction that suppresses non-memorized generations; the resulting adjusted prediction (dark green arrow) steers sampling toward memorized images from $\mathcal{D}_B$ and thus improves attack success.

Formally, MFG can be expressed as,

$$\epsilon_{\text{MFG}}(x_t, t) = \epsilon_{\theta_g^{r_2}}(x_t, t) + \gamma \cdot \left( \epsilon_{\theta_g^{r_2}}(x_t, t) - \epsilon_{\theta_g^{r_1}}(x_t, t) \right), \tag{2}$$

where $x_t$ denotes the noisy image at timestep $t$, $\epsilon_{\theta_g^{r_2}}(x_t, t)$ and $\epsilon_{\theta_g^{r_1}}(x_t, t)$ are predictions from the later-stage and earlier-stage global models, respectively. The scalar $\gamma \geq 0$ controls the strength of memorization-focused correction. The resulting noise prediction $\epsilon_{\text{MFG}}(x_t, t)$, shown as the dark green arrow in Fig. 3 (b), effectively biases the sampling trajectory toward reproducing memorized private images from $\mathcal{D}_B$.

Table 1: $\text{Mem}_\alpha$ (%) results of different data stealing methods on AFHQ-Dog (Choi et al., 2020) and CelebA (Liu et al., 2015), evaluated at $\alpha = 0.1$ and $\alpha = 0.2$. Baseline indicates generating images *without* TFG and MFG.

| Method | AFHQ-Dog (Choi et al., 2020) | | CelebA (Liu et al., 2015) | |
|---|---|---|---|---|
| | $\text{Mem}_{0.1}$ | $\text{Mem}_{0.2}$ | $\text{Mem}_{0.1}$ | $\text{Mem}_{0.2}$ |
| **Inversion-Based Attacks** | | | | |
| GIFD (Fang et al., 2023) | 0.00 | 0.00 | 0.00 | 0.00 |
| FedInverse (Wu et al., 2024) | 0.00 | 0.00 | 0.00 | 0.00 |
| **Memorization-Based Attacks (Ours)** | | | | |
| Baseline | 4.14 | 14.82 | 13.20 | 29.85 |
| Ours w/o TFG | 13.69 (+9.55) | 21.72 (+6.90) | 28.55 (+15.35) | 42.95 (+13.10) |
| Ours w/o MFG | 20.31 (+16.17) | 39.00 (+24.18) | 35.50 (+22.30) | 50.80 (+20.95) |
| Ours | 25.02 (+20.88) | 41.85 (+27.03) | 38.25 (+25.05) | 52.18 (+22.33) |

Built on TFG and MFG, we derive the complete data stealing procedure as shown in Algorithm 1. At each denoising timestep, we first apply TFG to the global-model predictions at both rounds $r_1$ and $r_2$—yielding $\epsilon_{\text{TFG}}^{r_1}$ and $\epsilon_{\text{TFG}}^{r_2}$ that suppress generations aligned with the attacker's own data—and then fuse these two TFG-adjusted predictions via MFG to produce the final noise prediction $\epsilon_{\text{MFG}}$ used for the denoising step ($x_t$ and $t$ are eliminated for clarity). Under this procedure, an attacker can maximize recovery of other participants' private images entirely after training completes.

## 4 EXPERIMENTS

### 4.1 EXPERIMENTAL SETTINGS

**Datasets.** We conduct experiments on widely adopted CelebA (Liu et al., 2015) and AFHQ (Choi et al., 2020) datasets to evaluate the effectiveness of our proposed memorization-guided attack.

**Evaluation Metric.** To quantify how many unique private training images have been successfully recovered, we introduce **$\text{Mem}_\alpha$**, a memorization metric based on Carlini's detection rule (Carlini et al., 2023). An image $x$ generated by the attacker is considered to memorize a training image $\hat{x} \in \mathcal{D}_B$ if it satisfies a relative distance threshold,

$$\ell(x, \hat{x}; S_x^n) = \frac{\ell_2(x, \hat{x})}{\mathbb{E}_{y \in S_x^n}[\ell_2(x, y)]} \leq \alpha, \tag{3}$$

where $S_x^n$ is the set of $n$ nearest neighbors of $x$ in the $\mathcal{D}_B$ under $\ell_2$ distance. However, we observe that multiple generated samples may correspond to the same memorized training image, overestimating the effective number of stolen instances. To address this, we define $\text{Mem}_\alpha$ as the ratio of unique images in $\mathcal{D}_{\text{benign}}$ that are identified as memorized by at least one generated sample,

$$\text{Mem}_\alpha = \frac{|\{\hat{x} \in \mathcal{D}_B \mid \exists x \in \mathcal{I}, \ell(x, \hat{x}; S_x^n) \leq \alpha\}|}{|\mathcal{D}_B|}. \tag{4}$$

This metric provides a clear measure of how many unique private images have been stolen, a smaller $\alpha$ indicates a stricter criterion for determining memorized samples.

**Baselines.** We adopt inversion-based methods GIFD (Fang et al., 2023) and FedInverse (Wu et al., 2024) as baselines. These methods require extra information (e.g., gradients or noise) to perform client-to-client attacks, which is unavailable in our setting. We provide such inputs to ensure feasibility, whereas our method **requires no additional information**, demonstrating its practicality and effectiveness.

**Implementation Details.** We adopt the training protocol from (Gan et al., 2024) to implement federated diffusion models. To prevent overfitting-induced memorization, all training is terminated once the model converges to a desirable FID on the test set. We assume a simplified attack scenario where only one client is malicious. By default, we report the results when client number is 5. All experiments are conducted on a workstation equipped with 8 NVIDIA GeForce RTX 4090 GPUs.

Table 2: Ablation on $r_2$ and $r_1$ selection in MFG. We report $\text{Mem}_{0.1}$/$\text{Mem}_{0.2}$ (%) on AFHQ-Dog and CelebA. $r_1$ and $r_2$ are expressed as percentages of the final global round $R$. Guidance scales $\beta, \gamma$ are fixed as 0.5.

| | Varying $r_2$ ($r_1 = 0.20 \times R$) | | | | | Varying $r_1$ ($r_2 = R$) | | | |
| --- | --- | --- | --- | --- | --- | --- | --- | --- | --- |
| $r_2$ | AFHQ-Dog | | CelebA | | $r_1$ | AFHQ-Dog | | CelebA | |
| | $\text{Mem}_{0.1}$ | $\text{Mem}_{0.2}$ | $\text{Mem}_{0.1}$ | $\text{Mem}_{0.2}$ | | $\text{Mem}_{0.1}$ | $\text{Mem}_{0.2}$ | $\text{Mem}_{0.1}$ | $\text{Mem}_{0.2}$ |
| $0.60 \times R$ | 6.66 | 23.30 | 25.33 | 35.10 | $0.05 \times R$ | 29.06 | 39.40 | 32.25 | 45.20 |
| $0.70 \times R$ | 13.01 | 29.16 | 30.04 | 44.78 | $0.10 \times R$ | 31.83 | 43.59 | 40.13 | 54.03 |
| $0.80 \times R$ | 16.64 | 32.91 | 35.25 | 52.38 | $0.15 \times R$ | 28.74 | 51.79 | 40.01 | 48.70 |
| $0.90 \times R$ | 23.04 | 39.35 | 31.53 | 46.85 | $0.20 \times R$ | 28.30 | 39.35 | 38.25 | 52.18 |
| $1.00 \times R$ | 25.02 | 41.85 | 38.25 | 52.18 | $0.40 \times R$ | 23.92 | 34.68 | 33.18 | 46.95 |

Table 3: Ablation on $\beta$ and $\gamma$ selection in TFG/MFG. We report $\text{Mem}_{0.1}$/$\text{Mem}_{0.2}$ (%) on AFHQ-Dog and CelebA. Global round $r_2$ is fixed as the final round $R$, while $r_1 = 0.2 \times R$.

| | Varying $\gamma$ ($\beta = 0.5$) | | | | | Varying $\beta$ ($\gamma = 0.5$) | | | |
| --- | --- | --- | --- | --- | --- | --- | --- | --- | --- |
| $\gamma$ | AFHQ-Dog | | CelebA | | $\beta$ | AFHQ-Dog | | CelebA | |
| | $\text{Mem}_{0.1}$ | $\text{Mem}_{0.2}$ | $\text{Mem}_{0.1}$ | $\text{Mem}_{0.2}$ | | $\text{Mem}_{0.1}$ | $\text{Mem}_{0.2}$ | $\text{Mem}_{0.1}$ | $\text{Mem}_{0.2}$ |
| 0.1 | 23.44 | 41.85 | 37.93 | 52.78 | 0.1 | 20.94 | 34.23 | 35.25 | 47.78 |
| 0.3 | 27.51 | 45.17 | 38.28 | 53.40 | 0.3 | 28.64 | 43.88 | 40.08 | 52.85 |
| 0.5 | 25.02 | 41.85 | 38.25 | 52.18 | 0.5 | 25.02 | 41.85 | 38.25 | 52.18 |
| 0.7 | 26.71 | 44.09 | 31.80 | 47.83 | 0.7 | 17.77 | 42.62 | 25.43 | 44.98 |
| 0.9 | 20.31 | 38.92 | 26.43 | 41.83 | 0.9 | 2.48 | 29.17 | 12.80 | 36.18 |

## 4.2 Main results

**Training Federated Diffusion Model.** We simulate a realistic FL setting with 5 clients jointly training a diffusion model, in order to faithfully evaluate the privacy risks posed by our method. To avoid overfitting that could artificially amplify memorization and bias our conclusions, we monitor the FID of the global diffusion model on an **unseen validation set** throughout training. As shown in Fig. 4 in Appendix, FID consistently decreases as training progresses, and training is terminated when the validation FID converges, ensuring that our analysis reflects genuine memorization rather than overfitting artifacts.

**Effectiveness of our method.** As shown in Tab. 1, the baseline without any guidance exposes only a very small fraction of training samples (e.g., 4.14% on AFHQ-Dog and 13.20% on CelebA at $\alpha = 0.1$), highlighting the difficulty of the attack. With our guidance strategies, recovery improves substantially, with TFG and MFG each boosting performance by around 10–20% on average, and their combination further raising recovery by over 25%. Unless otherwise specified, we set $\beta = \gamma = 0.5$, $r_2 = R$, and $r_1 = 0.2 \times R$, with rationale explained in Sec. 4.3. These results confirm that our method, especially with both guidance mechanisms, can recover a large fraction of benign clients' training data, posing a severe and realistic privacy threat.

**Comparisons with Inversion-based Attacks.** Inversion-based methods rely on information *unavailable* in our client-to-client setting (e.g., GIFD (Fang et al., 2023) requires victim gradients, FedInverse (Wu et al., 2024) needs added noise and timesteps during DDPM Ho et al. (2020) training). However, we still provided them with these private information, yet they failed to recover any private images (Tab. 1). In sharp contrast, our memorization-based attack, *without any extra information*, successfully reconstructs tens of percents of training data, clearly exposing the privacy risks in federated diffusion.

## 4.3 Ablation Study

We conduct an ablation study to examine how hyperparameters influence the attack success of our method. Specifically, we vary the choice of global models ($\theta_g^{r_1}, \theta_g^{r_2}$) and the guidance scales ($\beta, \gamma$) of TFG and MFG, and evaluate performance using $\text{Mem}_\alpha$.

Table 4: Performance of our method under various federated defense mechanisms. We report $Mem_{0.1}$ and $Mem_{0.2}$ scores on AFHQ-Dog and CelebA.

| Defense | AFHQ-Dog | CelebA |
|---|---|---|
| | $Mem_{0.1}$ / $Mem_{0.2}$ | $Mem_{0.1}$ / $Mem_{0.2}$ |
| Multi-Metrics | 16.15 / 29.40 | 26.15 / 35.27 |
| Multi-Krum | 14.08 / 31.41 | 25.91 / 36.96 |

Table 5: Attack performance under non-iid data distribution on AFHQ. We adopt $\alpha = 1.2$ to for Dirichlet distribution (Hsu et al., 2019) to split the dataset for all clients. We report $Mem_{0.1}$/$Mem_{0.2}$ (%).

| Setting | $Mem_{0.1}$ | $Mem_{0.2}$ |
|---|---|---|
| IID | 31.83 | 43.59 |
| Non-IID | 18.64 | 41.50 |

**Effect of later-stage global model selection.** Tab. 2 (left) shows that using later-stage $r_2$ consistently improves attack success when $r_1$ is fixed. This aligns with the finding of (Gu et al., 2025) that diffusion models memorize more training samples as training progresses. Consequently, selecting a larger $r_2$ allows the attacker to exploit these stronger memorization potential, leading to the recovery of more private images.

**Effect of earlier-stage global model selection.** As shown in Tab. 2 (right), varying $r_1$ with $r_2$ fixed produces a rise-then-fall trend. In the very early training stages, models generate outputs that resemble meaningless noise; incorporating such $r_1$ does not provide the non-memorized distribution needed for MFG and instead disrupts the predictions of $r_2$. As training progresses, earlier-stage models begin to capture general semantic features of the training data while still avoiding memorization of specific samples. This provides the most effective contrast for MFG, leading to the highest attack success. However, once $r_1$ itself starts memorizing specific samples, it no longer represents the non-memorized distribution, and its guidance becomes less effective, causing performance to decline. This observation aligns well with the principle of MFG, which relies on contrasting memorized and non-memorized generations.

**Effect of guidance scales.** Tab. 3 presents the results of two guidance scales. We observe that both AFHQ-Dog and CelebA exhibit similar optimal ranges, with $\beta$ and $\gamma$ performing best between $0.3$ and $0.7$. This indicates that the choice of guidance strength is relatively stable across datasets, and practical attacks can adopt these values as default settings.

### 4.4 ROBUSTNESS OF ANALYSIS

**Influence of defense mechanisms.** As shown in Tab. 4, we evaluate the robustness of our method under two mainstream federated defense mechanisms, including, Multi-Metrics (Huang et al., 2023), and Multi-Krum (Blanchard et al., 2017). The result shows that our method remains effective under federated defense mechanisms.

**Effectiveness of our method under non-iid settings.** We sample 5,000 images uniformly from the three categories of AFHQ and partition them across 5 clients using a Dirichlet distribution (Hsu et al., 2019) with $\alpha = 1.2$ to simulate a non-iid training setup. As shown in Tab. 5, our method still successfully recovers 18.64% of training data, demonstrating its robustness under non-iid settings.

## 5 CONCLUSIONS

This work reveals a realistic client-to-client privacy leakage risk in federated diffusion models, where memorized training images can be regenerated by malicious clients. To expose and study this threat, we propose memorization-guided attack, which involves two generation guidance, TFG and MFG, to maximize the data stealing performance. Extensive experiments demonstrate that tens of percents of private images can be stolen by our method under this realistic scenario, highlighting the urgent need for stronger privacy protections in federated generative learning.

## 6    ETHICS STATEMENT

This work investigates the privacy risks of federated diffusion models by introducing a client-to-client data stealing attack. Our study does not involve human subjects, and all experiments are conducted on publicly available datasets (AFHQ (Choi et al., 2020) and CelebA (Liu et al., 2015)) under standard research licenses. We explicitly avoid releasing any potentially sensitive or private data and restrict our implementation to controlled experimental settings. The purpose of this work is not to enable malicious attacks, but rather to highlight critical vulnerabilities in federated diffusion frameworks and motivate the development of stronger defenses. We believe that raising awareness of such threats is essential for the community to design secure, privacy-preserving generative modeling systems in compliance with ethical and legal standards.

## 7    REPRODUCIBILITY STATEMENT

We have taken extensive measures to ensure the reproducibility of our work. All implementation details of the proposed attack, including training configurations, hyperparameter choices, and guidance scales, are thoroughly described in Sec. 4 and Sec. A.1. We also specify dataset usage , preprocessing steps, and evaluation metrics to allow precise replication of our experimental results. In the supplementary material, we provide additional implementation notes for guidance. Together, these resources are intended to make it straightforward for researchers to reproduce and verify our findings.

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

# A APPENDIX

## A.1 MORE IMPLEMENTATION DETAILS

### A.1.1 INVERSION-BASED BASELINES

**GIFD Fang et al. (2023).** We use the official implementation of GIFD[1] and adapt it to the diffusion inversion setting. Unlike classification models where gradients are computed with respect to the input directly, the gradient of a diffusion model depends jointly on the clean image $x_0$, the added noise $\epsilon$, and the sampled timestep $t$. Since all three variables are typically unknown to the attacker, performing triplet optimization over $(x_0, \epsilon, t)$ is highly ill-posed and computationally infeasible. To make the optimization tractable, we follow the assumption proposed by GIDM Huang et al. (2024a) that the attacker has access to the values of $\epsilon$ and $t$, and performs gradient-based optimization only with respect to $x_0$ while keeping $\epsilon$ and $t$ fixed. For pre-trained GANs, we adopt BigGAN Brock et al. (2019) trained on ImageNet Deng et al. (2009) for inversion on the AFHQ-Dog dataset, and StyleGAN2 Karras et al. (2020) trained on FFHQ Karras et al. (2019) for inversion on the CelebA dataset. For BigGAN, we optimize all 13 layers for 500 steps per layer with a learning rate of 0.1. For StyleGAN2, we similarly optimize all 8 layers for 500 steps each with the same learning rate. All other hyperparameters follow the official implementations.

**FedInverse Wu et al. (2024).** We follow the official implementation of FedInverse[2] and adopt its GMI variant for inversion. Unlike classification models that provide semantically meaningful class logits to guide the inversion process, diffusion models do not offer such direct supervision. To address this gap, we record the random noise $\epsilon$ and the corresponding timestep $t$ used during training, and treat each $(\epsilon, t)$ pair as a pseudo-class label to guide the inversion. All other inversion-related hyperparameters are kept consistent with the original implementation.

### A.1.2 FEDERATED DIFFUSION

**Denoising Unet.** We follow the widely adopted DDPM Ho et al. (2020) implementation from the repository[3] to construct our denoising model, which adopts a symmetric U-Net architecture composed of a series of residual blocks with skip connections between encoder and decoder stages. The network begins with a $3 \times 3$ convolutional layer projecting the input to the base channel dimension of 128. It then traverses five resolution levels, with channel multipliers set to $[1, 2, 2, 2, 4]$, resulting in progressively wider feature representations. At each resolution level, we stack two residual blocks, each consisting of a GroupNorm Wu & He (2018) layer with 32 groups, a Swish activation Ramachandran et al. (2017), a $3 \times 3$ convolution, and additive modulation from the time-step embedding. Between the encoder and decoder, the architecture includes two additional residual blocks operating at the lowest resolution, which serve as the middle blocks of the network. These blocks have the same structure as standard residual blocks but do not include attention in our configuration. The time-step embedding is constructed by applying sinusoidal positional encoding to discrete timestep indices, followed by two linear layers with Swish activation. Downsampling across resolution levels is implemented via strided $3 \times 3$ convolutions, while upsampling is performed using nearest-neighbor interpolation followed by $3 \times 3$ convolutions. In the decoder, feature maps from each stage are concatenated with the corresponding encoder features via skip connections before passing through the residual blocks. The final output is produced through a GroupNorm, Swish activation, and a $3 \times 3$ convolutional layer

**Training details.** During federated training, each participating client performs 100 epochs of local training on its private dataset using the denoising U-Net described above, after which model parameters are uploaded to the server for global aggregation. During aggregation, we employ standard FedAvg McMahan et al. (2017) to compute the average of the model weights across clients. The updated global model is then broadcast back to all clients to initiate the next communication round. For local training, we use the Adam optimizer with a learning rate of $1 \times 10^{-4}$, combined with a linear warmup schedule implemented via a LambdaLR scheduler. Gradient clipping is applied with a maximum norm of 1.0. We resize the input images into $64 \times 64$ and apply normalization

---

[1]https://github.com/ffhibnese/GIFD_Gradient_Inversion_Attack
[2]https://github.com/Jun-B0518/FedInverse/tree/main/GMI
[3]https://github.com/w86763777/pytorch-ddpm

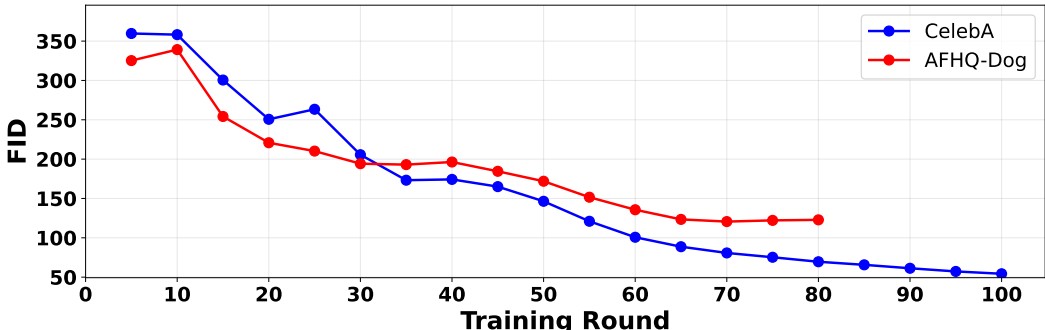

Figure 4: **Validation-set FID across FL training rounds on CelebA and AFHQ-Dog**. We report the FID on *unseen validation* data to track how the generative quality evolves during federated training. To mitigate overfitting and unintended memorization, training is stopped once the FID ceases to decrease significantly.

Table 6: Influence of aggregation frequency on attack performance under a fixed total training budget of 6,000 epochs for each clients. Aggregation frequency refers to how often the central server aggregates local models from clients and redistributes the global model in the federated learning process. We report $\text{Mem}_{0.1}(\%)$ and $\text{Mem}_{0.2}(\%)$ on AFHQ-Dog under the 5-client setting.

| Local Epochs | Global Rounds | w/o Ours | | w/ Ours | |
|---|---|---|---|---|---|
| | | $\text{Mem}_{0.1}$ | $\text{Mem}_{0.2}$ | $\text{Mem}_{0.1}$ | $\text{Mem}_{0.2}$ |
| 50 | 120 | 12.14 | 27.59 | 35.44 (+23.30) | 49.53 (+21.94) |
| 100 | 60 | 4.10 | 14.63 | 18.30 (+14.20) | 19.53 (+4.90) |
| 200 | 30 | 0.61 | 4.01 | 4.15 (+3.54) | 12.21 (+8.20) |

augmentation for all models. The denoising objective follows the original DDPM Ho et al. (2020) formulation, where the model learns to predict additive Gaussian noise under a fixed variance schedule. The noise schedule is defined by linearly interpolating $\beta_t$ values from $1 \times 10^{-4}$ to $2 \times 10^{-2}$ over 1000 timesteps. By default, we randomly sample 5,000 images from CelebA as the training set, while for AFHQ-Dog, we use the full training set. All training process terminate when validation set FID converge. FID is evaluated on val split of AFHQ-Dog and CelebA, which are unseen for all the clients.

**Sampling details.** We adopt the deterministic DDIM Song et al. (2021) sampling strategy for image generation, using a total of 50 denoising steps per image. We apply the guidance strategies proposed in Sec.3 across all the sampling steps.

## A.2 INFLUENCE OF CLIENT NUMBER

We investigate influence of the number of clients on attack performance. Results can be seen in Tab. 7. We report $\text{Mem}_{0.1}$ and $\text{Mem}_{0.2}$ under both w/ Ours and w/o Ours settings. Across all client configurations (5, 7, and 10), the introduction of our method consistently leads to substantial improvements. Specifically, under the 5-client setting, our method achieves a gain of +13.93 in $\text{Mem}_{0.1}$ and +19.66 in $\text{Mem}_{0.2}$. Similar trends are observed for 7 and 10 clients, demonstrating that our method remains effective regardless of the degree of data decentralization. These results confirm the robustness of our proposed method and highlight its ability to enhance attack performance under various scenarios.

## A.3 INFLUENCE OF AGGREGATION FREQUENCY

To assess the impact of aggregation frequency on the effectiveness of our proposed attack, we vary the number of local training epochs per communication round while keeping the total number of

Table 7: Influence of client number on attack performance. We report $\text{Mem}_{0.1}(\%)$ and $\text{Mem}_{0.2}(\%)$ on AFHQ-Dog. We fix the total training epochs as 6,000 for each client.

| Client Num. | w/o Ours | | w/ Ours | |
|---|---|---|---|---|
| | $\text{Mem}_{0.1}$ | $\text{Mem}_{0.2}$ | $\text{Mem}_{0.1}$ | $\text{Mem}_{0.2}$ |
| 5 | 4.10 | 14.63 | 18.03 (+13.93) | 34.29 (+19.66) |
| 7 | 5.39 | 16.17 | 19.29 (+13.90) | 36.87 (+20.70) |
| 10 | 3.21 | 13.15 | 9.47 (+6.26) | 30.87 (+17.72) |

Table 8: Effect of training set size on memorization recovery. We report $\text{Mem}_{0.1}/\text{Mem}_{0.2}$ (%) on **CelebA** under different numbers of training samples. Default attack hyperparameters: $(\beta, \gamma, r_1, r_2) = (0.5, 0.5, 0.2 \times R, R)$, where $R$ is the final round of FL training.

| Metric | 1K | 3K | 5K | 8K | 10K |
|---|---|---|---|---|---|
| $\text{Mem}_{0.1}$ | 87.22 | 66.93 | 38.25 | 14.25 | 5.28 |
| $\text{Mem}_{0.2}$ | 93.33 | 78.26 | 52.18 | 24.60 | 13.44 |

training epochs fixed at 6,000. As shown in Table 6, we observe that the frequency of aggregation has a clear influence on the baseline model's inherent memorization ability—lower aggregation frequency (i.e., more local updates per round) generally leads to weaker memorization and hence lower attack performance. Nevertheless, across all settings, our method consistently improves the amount of data successfully extracted, even when the underlying model exhibits limited memorization. This demonstrates that our method is robust and effective under varying communication schedules, and can reliably exploit available memorized signals regardless of their strength.

### A.4 IMPACT OF TRAINING SET SIZE

Since diffusion-model memorization is strongly dependent on dataset size: as documented in Fig. 1 of (Gu et al., 2025), models trained on larger datasets exhibit substantially less tendency to reproduce training samples. Consequently, our method, which explicitly exploits memorization, is inevitably sensitive to the number of training examples as shown in Tab. 8. Despite this sensitivity, our method is still the only feasible client-to-client attack under the realistic scenario, which highlights that federated diffusion systems can still pose a non-trivial privacy risk in realistic settings.

### A.5 DERIVING THREAT-FOCUSED GUIDANCE (TFG)

As discussed in Sec. 3.2.1, TFG aims to *suppress* the attacker's private distribution while *emphasizing* the global model's distribution so that sampling is biased toward images in $\mathcal{D}_B$. Following (Karras et al., 2024), the global model at round $r$ with parameters $\theta_g^r$ can be viewed as approximating the score $\nabla_x \log p_{g^r}(x; \sigma)$ of the noisy density $p_{g^r}(x; \sigma)$ at noise level $\sigma$, where $x$ has been corrupted by Gaussian noise $\mathcal{N}(0, \sigma^2 \mathbf{I})$. Consequently, sampling based solely on $\theta_g^r$ can be formalized as the score-based update

$$D_{g^r}(x; \sigma) \approx x + \sigma^2 \nabla_x \log p_{g^r}(x; \sigma), \tag{5}$$

where $D_{g^r}(\cdot; \sigma)$ denotes the denoising/drift update at noise standard deviation $\sigma$.

To suppress the private model during sampling, we *construct an extrapolation between the two score functions*,

$$s_{\text{TFG}}(x_t, t) = (1 + \beta) s_{g^r}(x_t, t) - \beta s_p(x_t, t), \qquad s_\phi(x_t, t) := \nabla_{x_t} \log p_\phi(x_t), \tag{6}$$

which *amplifies* the global score and *suppresses* the private score. Replacing the original score in Eq. 5 with the extrapolated score in Eq. 6—while keeping the diffusion coefficient $\sigma^2$ unchanged so that the per-step noise variance is preserved—yields the TFG denoising/drift update

$$D_{\text{TFG}}(x; \sigma) \approx x + \sigma^2 s_{\text{TFG}}(x; \sigma) = x + \sigma^2 \Big( (1 + \beta) s_{g^r}(x; \sigma) - \beta s_p(x; \sigma) \Big). \tag{7}$$

By the score–denoiser equivalence (Vincent, 2011),

$$s_\phi(x_t, t) \approx -\frac{1}{\sigma_t} \epsilon_\phi(x_t, t), \tag{8}$$

with $\sigma_t$ denoting the noise standard deviation at step $t$, we obtain the denoiser form

$$\epsilon_{\text{TFG}}(x_t, t) = (1 + \beta) \epsilon_{\theta_g^r}(x_t, t) - \beta \epsilon_{\theta_p}(x_t, t)$$
$$= \epsilon_{\theta_g^r}(x_t, t) + \beta (\epsilon_{\theta_g^r}(x_t, t) - \epsilon_{\theta_p}(x_t, t)), \tag{9}$$

which matches Eq. 1 in the main text. In practice, $\sigma_t$ is discretized over timesteps, consistent with DDIM (Song et al., 2021).

### A.6 DERIVING MEMORIZATION-FOCUSED GUIDANCE (MFG)

As discussed in Sec. 3.2.2, MFG aims to *down-weight* the distribution of an early-round global model while *emphasizing* the distribution of a later-round global model. Following Karras et al. (2024), the global model at round $r_j$ with parameters $\theta_g^{r_j}$ approximates the score $\nabla_x \log p_{r_j}(x; \sigma)$ of the noisy density $p_{r_j}(x; \sigma)$ at noise level $\sigma$, where $x \sim \mathcal{N}(0, \sigma^2 \mathbf{I})$ corruption. Hence, sampling based solely on the later-round model $\theta_g^{r_2}$ can be written as

$$D_{r_2}(x; \sigma) \approx x + \sigma^2 \nabla_x \log p_{r_2}(x; \sigma), \tag{10}$$

with $D_{r_2}(\cdot; \sigma)$ the denoising/drift update at noise standard deviation $\sigma$. To suppress the earlier-round distribution $p_{r_1}$ ($r_2 > r_1$) while promoting $p_{r_2}$, we *construct an extrapolation between the two score functions*

$$s_{\text{MFG}}(x_t, t) = (1 + \gamma) s_{r_2}(x_t, t) - \gamma s_{r_1}(x_t, t), \qquad s_\phi(x_t, t) := \nabla_{x_t} \log p_\phi(x_t), \tag{11}$$

which *amplifies* the later-round score and *suppresses* the early-round score. Replacing the score in Eq. 10 with $s_{\text{MFG}}$—while keeping the diffusion coefficient $\sigma^2$ unchanged so that the per-step noise variance is preserved—yields the MFG denoising/drift update

$$D_{\text{MFG}}(x; \sigma) \approx x + \sigma^2 s_{\text{MFG}}(x; \sigma) = x + \sigma^2 \big((1 + \gamma) s_{r_2}(x; \sigma) - \gamma s_{r_1}(x; \sigma)\big). \tag{12}$$

By the score–denoiser equivalence (Vincent, 2011),

$$s_\phi(x_t, t) \approx -\frac{1}{\sigma_t} \epsilon_\phi(x_t, t), \tag{13}$$

with $\sigma_t$ the noise standard deviation at step $t$, we obtain the denoiser form

$$\epsilon_{\text{MFG}}(x_t, t) = (1 + \gamma) \epsilon_{\theta_g^{r_2}}(x_t, t) - \gamma \epsilon_{\theta_g^{r_1}}(x_t, t)$$
$$= \epsilon_{\theta_g^{r_2}}(x_t, t) + \gamma (\epsilon_{\theta_g^{r_2}}(x_t, t) - \epsilon_{\theta_g^{r_1}}(x_t, t)), \tag{14}$$

which coincides with Eq. 2 in the main text. In practice, $\sigma_t$ is discretized over timesteps, consistent with DDIM (Song et al., 2021).

### A.7 LLM USAGE

We used large language models (LLMs) solely as writing assistants to polish the presentation and improve the clarity of our manuscript. LLMs were not involved in research ideation, experimental design, implementation, analysis, or any other scientific contribution. All technical content, methodology, results, and conclusions were fully conceived, developed, and validated by the authors. The authors take full responsibility for the correctness and integrity of the content.

### A.8 MORE VISUALIZATION RESULTS

We provide more visualization of the training images and their corresponding stolen images from AFHQ-Dog and CelebA. Visualization results can be seen below.

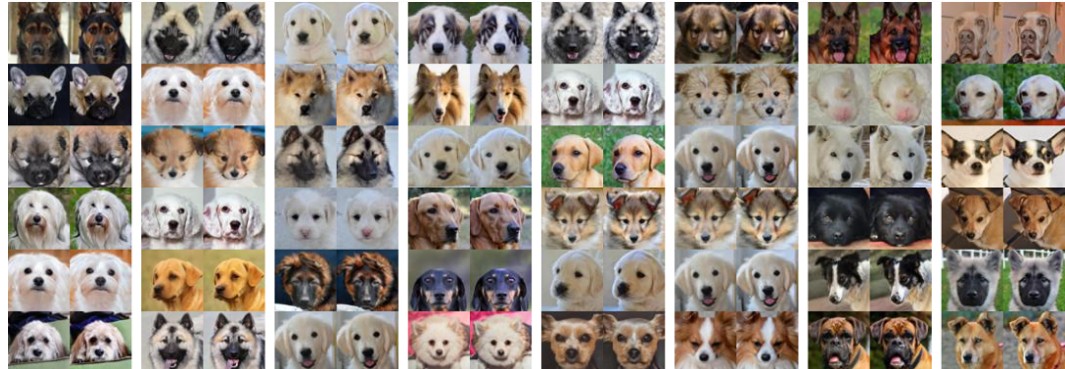

Figure 5: The grid image of the training and stolen images of AFHQ-Dog. Odd-numbered columns show training images, and even-numbered columns show the closest stolen samples in training set.

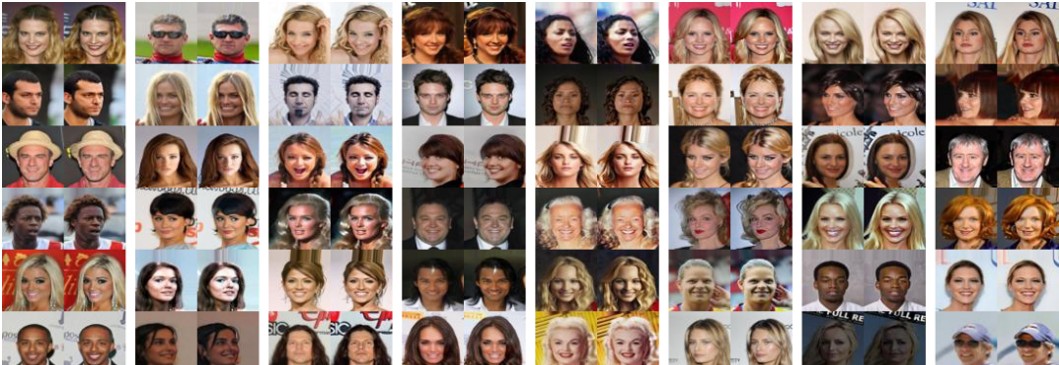

Figure 6: The grid image of the training and stolen images of CelebA. Odd-numbered columns show training images, and even-numbered columns show the closest stolen samples in training set.

