# OpenReview forum: "Mining Your Memory: Client-to-Client Data Stealing in Federated Diffusion Model through Memorization"
_ICLR.cc/2026/Conference — ICLR 2026 Conference Withdrawn Submission_

### Official Review · Reviewer_URQY · 2025-10-27

**Soundness:** 3
**Presentation:** 3
**Contribution:** 2
**Rating:** 4
**Confidence:** 3

**Summary:**

This paper studies a novel and practically relevant privacy threat in federated diffusion models (FDMs): a client-to-client data stealing attack conducted after training by a curious client using only the publicly shared global model snapshots. The attack leverages the inherent memorization property of diffusion models. The authors propose two guidance mechanisms: (1) Threat-Focused Guidance (TFG): Steers the sampling away from the attacker's own data distribution using a privately trained local model. (2) Memorization-Focused Guidance (MFG): Amplifies memorized samples by comparing early-round versus late-round global models. The combined guidance enables the malicious client to generate samples that closely replicate private training images held by other clients. The paper reports up to 25–52% recovery rate (Memα metric) under realistic settings on federated CelebA and AFHQ-Dog datasets. The authors further test robustness under non-IID distributions and Byzantine-robust aggregation.

**Strengths:**

1. Novel threat model. Shifts focus from server-side/gradient inversion attacks to client-side post-hoc generative extraction, matching real-world FDM deployments.
2. Simple yet effective methodology. The guidance mechanism is lightweight, requires no training modifications, and is easily implementable during inference.
3. Clear writing and reproducibility. Algorithm, formulas, and training specifics are carefully documented.

**Weaknesses:**

1. Potentially trivial threat model. Although the threat model is novel, it raises concerns regarding whether a separate client-side extraction algorithm is truly necessary. A server could also extract private data from individual gradient updates using [1]. The proposed client-side extraction method targets extracting private information from all other clients collectively rather than from a specific target client. It seems that server-side extraction methods are capable of achieving the same objective. The authors should clarify why the proposed client-side extraction is not trivially reducible to server-side extraction methods.

2. Missing baselines from existing data extraction methods. The authors only compare their approach against two extraction baselines, without including [1], from which they draw their evaluation metrics. Providing empirical results using existing data extraction algorithms would significantly strengthen the contribution.

3. Limited defense evaluation. The authors only evaluate against two basic anomaly detection mechanisms in federated learning, but do not consider differential privacy (which is widely adopted for privacy protection) or other defenses specifically designed for data extraction attacks.

[1] Extracting Training Data from Diffusion Models. USENIX Security 2023. Carlini.

**Questions:**

1. Could authors clarify the necessity of the specific client-side data extraction methods? Why existing server side methods could not be just applied to client side?

2. Could authors provides comparsion with more data extraction methods?

3. Could authors provides evaluations against differential privacy or other defense methods?

---

### Official Review · Reviewer_qZAK · 2025-10-30

**Soundness:** 2
**Presentation:** 3
**Contribution:** 1
**Rating:** 4
**Confidence:** 4

**Summary:**

The paper proposed a guided solution using malicious clients in federated diffusion where they can guide the global model to generate images of memorized training data from other victim models. Experiments are provided to support the proposed attack.

**Strengths:**

1. One important contribution of the paper is the metric mem-alpha which uses Carlini rules to define whether an image is memorized.

**Weaknesses:**

1. Gan et al paper is briefly mentioned in related work saying they neglect easy-to-memorize images. However, their settings are the same as this paper's setting and the proposed attack is also the same where malicious clients participate to guide the global memorize and generate private training images of other clients in federated learning. Can authors elaborate on what the difference is? How do you contribute to the novelty and to this research question? Also, why [1] is not in the baselines for evaluation?

2. It is common to use malicious clients to guide the global model leak memorized training samples of victim clients in federated learning. The question has been well explored under settings like CNN, etc. The reviewer believes there would be a contribution to putting such a question into federated diffusion. However, the methods are following the same idea as the previous methods. Can authors elaborate on why this is the major contribution of this paper apart from bringing such a question into federated diffusion?

3. The framework is based on full-model tuning which is not common in today's large foundation model federated learning. Why don't you study the case where LoRA matrices are uploaded. LoRA matrices are basically some linear computation. The malicious injection can be mitigated and there have been methods to mitigate it. [2]

4. The algorithm relies on deviations of two model gradients and guides the global model to memorize other client model data. The reason is that malicious client gradients are easy to differentiate. So, the remaining gradients should be resolved from the server. Correct me if I understand it wrongly. The server (or other clients) can easily detect this if they either run inference on the received model or check the gradients. My question is how does the proposed solution pass such kinds of detection?

5.`We simulate a realistic FL setting with 5 clients jointly training a diffusion model. This setting is far from realistic. Even for cross-silo case, there are more clients participated in the learning phase.` Five clients in real-world can easily meet the case that everyone is down due to the connection and other issues. Besides, it is barely possible to connect enough samples from only 5 clients. CelebA has 200K+ samples which means each client at least has 40K+ samples which is far from real world cases if samples are averagely distributed. The local datacenter should be at a certain scale to have the possibility of collecting so many samples, let alone that they may be all sensitive data.

[1] Gan, Yuan, Jiaxu Miao, and Yi Yang. "DataStealing: Steal Data from Diffusion Models in Federated Learning with Multiple Trojans." Advances in Neural Information Processing Systems 37 (2024): 132614-132646.
[2] Bossy, Thierry, Julien Vignoud, Tahseen Rabbani, Juan R. Troncoso Pastoriza, and Martin Jaggi. "Mitigating Unintended Memorization with LoRA in Federated Learning for LLMs." arXiv preprint arXiv:2502.05087 (2025).

**Questions:**

Please see my questions in weakness point 1, 3, and 4.

---

### Official Review · Reviewer_Zs4q · 2025-10-30

**Soundness:** 1
**Presentation:** 3
**Contribution:** 2
**Rating:** 2
**Confidence:** 4

**Summary:**

This work introduces a new kind of data stealing attack in a federated diffusion training setup. The adversary is a single malicious client that take advantage of diffusion models vulnerability - training images memorization - to exploit collaborative training process and obtain images contributed by other clients. The method is based on generative model guidance having two aims: 1) MFG - to increase memorization 2) TFG - to increase the probability of generating data from other clients rather than from adversarial client. The authors alert that explicit data isolation does not truly eliminate privacy leakage, as  even under normal FL circumstances, with no additional information the malicious client could get access to memorized samples from other clients through generations of the final trained model.

**Strengths:**

- The memorization problem is already well known, but the broad consequences of this phenomenon are still understudied. This work shows dangers hidden in collaborative training of a diffusion model for client-to-client stealing, which is a novel adaptation of a known issue and can be interesting.
- the attack is not using any additional information, the adversary has only information that every client gets during the training
- the main idea is clearly stated and explained, the writing is easy to follow,
- interesting ablation experiments are added,
- the results seem to show that the threat is really alerting

**Weaknesses:**

W1.
FID results shown on fig. 4 (in the appendix, not in the main text!) may be a red flag for evaluation of the proposed method. It shows that the final generation quality of the model is bad and undermine the presented results (all reported results in the main part use only a metric introduced by the authors). The qualitative results are not shown and the quality of generation can only be inferred from very high FID results.

W2.
Following W1, proposed metric may not be a good way to quantify the threat of data stealing in this setup - the chance of stealing real good quality image with bad generations are low. It is the only metric used.

W3.
Memorization mitigation strategies were listed, but the authors did not use any memorization mitigation strategy to test if it is a valid defense. Additionally, the model is trained (or fine-tuned?) on a very small subsets of data (1000 images x5) which is only shown in the appendix - it is highly possible that a big model with small number of data sample will have a huge memorization, but the vulnerability is much less threatening with only a little more data for training. The threat with 2000 x5 is already much lower (table in the appendix).

W4.
Other:
- no possible defense approaches against a novel attack is even mentioned or discussed - but the attack is indeed tested under two FL defenses from before the memorization problem was known. The authors claim that increasing awareness  of threats is essential, but are focused narrowly on stealing attack rather than employing the same methods for defensive purposes which is harder, but possible.
- no method limitations discussed
- many important issues in the appendix, not stated clearly in the main text.
- table 2 - right: results for 0x20 R = r1 must be a mistake, it is not consistent with the rest of the results.

**Questions:**

- weaknesses W1-W3
- W2. Mem % is only calculated as % of other clients data?
- W4. In principle any of the benign clients could use TFG guidance as well for defensive purposes and use MFG guidance in the opposite direction to prevent the memorization. Have you considered this simple setup and test if it mitigates the attack successfully?

---

### Official Review · Reviewer_NgZS · 2025-10-31

**Soundness:** 3
**Presentation:** 2
**Contribution:** 2
**Rating:** 4
**Confidence:** 4

**Summary:**

The paper presents an attack framework that enables a client participating in federated training of a diffusion model to reconstruct images from other clients’ training data. The attack exploits the model’s memorization behavior and requires only access to intermediate and final model checkpoints shared through the standard federated protocol. No interference with the training process is assumed. The main idea is to compare predictions from an earlier, less-trained model (which has memorized less) to those of a later, more-trained model (which has memorized more). By differencing these predictions, the attacker can guide the diffusion model toward generating memorized training samples with higher likelihood.

**Strengths:**

+ The experiments are thorough and evaluate the proposed attack under realistic diffusion training conditions.

+ The core idea of guiding diffusion models via prediction differencing across training checkpoints appears novel and empirically effective. Figures 5 and 6 demonstrate high-quality reconstructions.

+ The visual examples of the attack’s recovered images provided in the appendix are impressive; reconstructions are very close to the original images and also high quality. I suggest that Figures 5 and 6 be moved to the main paper to provide a qualitative measure of the attack’s effectiveness.

**Weaknesses:**

- The paper reports the fraction of training images that appear within a similarity threshold among the generated samples but does not specify how many total images were generated to achieve this recovery rate. This information is critical since an attacker cannot distinguish genuine reconstructions from unrelated samples.

- Table 8 shows that increasing the number of training images significantly weakens the attack (e.g., recovery drops by >75% when moving from 5k to 10k samples). From the limited data shown, the degradation rate also seems superlinear. This raises concerns about the attack’s scalability, as diffusion training often involves far larger datasets where the attack might be ineffective. The paper should discuss the asymptotic behavior or practical limits of the attack with respect to the dataset size.

- The introduction and related work do not clarify whether the technique of guiding diffusion models via subtracting predictions is novel. The paper neither claims it explicitly nor cites prior works using similar methods, leaving the level of technical contribution ambiguous.

- The comparisons against FedInverse and GIFD seem inappropriate, as those methods target classifiers rather than generative models and are not able to reconstruct any training images in this attack setting. Since the paper also notes that unmodified diffusion models produce memorized images spontaneously, the unattacked model’s memorization rate would be a more meaningful baseline if no direct comparison exists.

**Questions:**

- What is the ratio of generated to recovered images? How many images must be generated to recover the percentages listed in the results section?

- Is the technique of differencing the predictions of multiple diffusion models, or the same model at different stages of training new, or has it been applied previously?

---

### Note · Authors · 2026-01-04

I have read and agree with the venue's withdrawal policy on behalf of myself and my co-authors.